# FRUSTRATINGLY EASY QUASI-MULTITASK LEARNING

## ABSTRACT

We propose the technique of quasi-multitask learning (Q-MTL), a simple and easy to implement modification of standard multitask learning, in which the tasks to be modeled are identical. We illustrate it through a series of sequence labeling experiments over a diverse set of languages, that applying Q-MTL consistently increases the generalization ability of the applied models. The proposed architecture can be regarded as a new regularization technique encouraging the model to develop an internal representation of the problem at hand that is beneficial to multiple output units of the classifier at the same time. This property hampers the convergence to such internal representations which are highly specific and tailored for a classifier with a particular set of parameters. Our experiments corroborate that by relying on the proposed algorithm, we can approximate the quality of an ensemble of classifiers at a fraction of computational resources required. Additionally, our results suggest that Q-MTL handles the presence of noisy training labels better than ensembles.

## 1 INTRODUCTION

Ensemble methods are frequently used in machine learning applications due to their tendency of increasing model performance. While the increase in the prediction performance is undoubtedly an important aspect when we train a model, it should not be forgotten that the increased performance of ensembling comes at the price of training multiple models for solving the same task.

The question that we tackle in this paper is the following: *Can we enjoy the benefits of ensemble learning, while avoiding its overhead for training models from scratch multiple times?* This question is highly relevant these days, since state-of-the-art neural models tend to be extremely resource-intensive on their own (Strubell et al., 2019), prohibiting their inclusion in a traditional ensemble setting.

Our proposed architecture simultaneously offers the benefit of ensemble learning, while avoiding its drawback of training multiple models. The method introduced here employs a special form of multitask learning (MTL). Caruana (1997) argues in his seminal work that MTL can be a useful source of introducing inductive bias into machine learning models. Standard MTL have been shown to be fruitfully applicable in solving a series of NLP tasks: Collobert & Weston (2008); Plank et al. (2016); Rei (2017); Kiperwasser & Ballesteros (2018); Sanh et al. (2018), *inter alia*. We introduce quasi-multitask learning (Q-MTL), where the goal is to simultaneously learn multiple neural models that solve *identical tasks*, while relying on a *shared representation* layer.

Besides the considerable speedup that comes with the proposed technique, we additionally argue that by applying multiple output units on top of a shared parameter set is beneficial, as we can avoid converging to such degenerate internal representations that are highly tailored for a particular classification model. In that sense, Q-MTL can also be viewed as an implicit regularizer, which prevents neural networks to develop such an internal representation which is not generic enough to provide useful input to multiple classification units simultaneously.

Our experiments with Q-MTL illustrate that the presence of multiple classifier layers for the same task affect each other positively – similar to ensemble learning – without the additional overhead of actually training multiple models.

A similar technique have already been derived from MTL called Pseudo-Task Augmentation (Meyerson & Miikkulainen, 2018), which builds on the idea of common representation, but the manage-

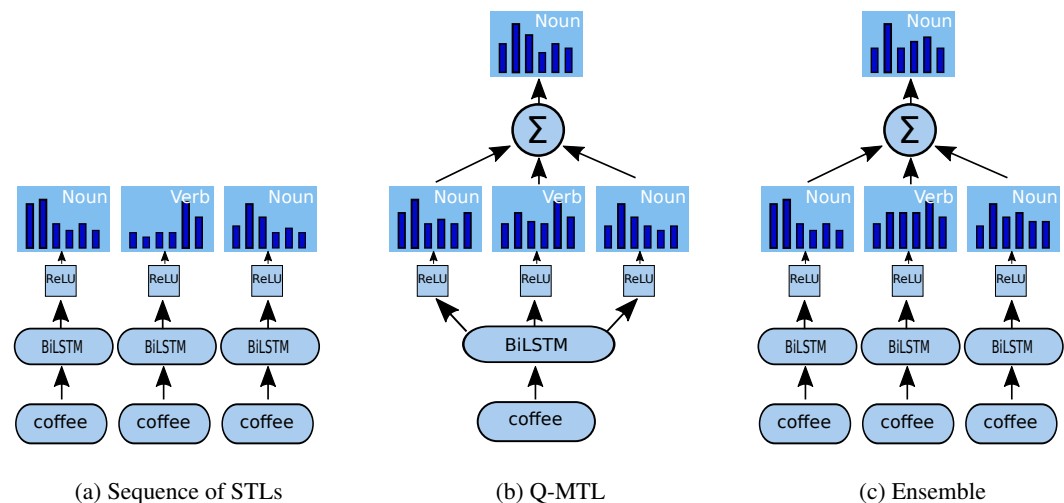

Figure 1: A schematic illustration of the different architectures employed in our experiments. Quasi-Multitask Learning (Q-MTL) averages the predictions of multiple classification units similar to ensembling without the computational bottleneck of adjusting the parameters of multiple LSTM cells.

ment of these tasks differs. We conducted experiments comparing the two methods for a greater comprehension of the differences.

## 2 APPLIED MODELS

We release all our source code used for our experiments at `anonymized`. Our models are based on the sequence classification framework from Plank et al. (2016) implemented in DyNet (Neubig et al., 2017). Figure 1 provides a visual summary of the different architectures we implemented. Figure 1b highlights that Q-MTL has the benefit of training multiple classification models over the same internal representation, as opposed to traditional ensemble model, which requires the training of multiple LSTM parameters as well (cf. Figure 1c).

### 2.1 BASELINE ARCHITECTURE

Our baseline classifier is a bidirectional LSTM (Hochreiter & Schmidhuber, 1997) incorporating character and word level embeddings. We first compute the input embedding for the network at position $i$ as

$$\mathbf{e_i} = \mathbf{w_i} \oplus \overrightarrow{\mathbf{c_i}} \oplus \overleftarrow{\mathbf{c_i}},$$

where $\oplus$ is the concatenation operator, $\mathbf{w_i}$ denotes the word embedding, $\overrightarrow{\mathbf{c_i}}$ and $\overleftarrow{\mathbf{c_i}}$ refers to the left-to-right and right-to-left character-based embeddings, respectively. We subsequently feed $\mathbf{e_i}$ into a bi-LSTM, which determines a hidden representation $\mathbf{h_i} \in \mathbb{R}^m$ for every token position as $\mathbf{h_i} = \overrightarrow{\mathbf{h_i}} \oplus \overleftarrow{\mathbf{h_i}}$, i.e., the concatenation of the hidden states of the two LSTMs processing the input from its beginning to the end, and in reverse direction.

The final output of the network for token position $i$ gets computed as

$$\mathbf{y_i} = softmax(ReLU(V\mathbf{h_i} + \mathbf{b_V})W + \mathbf{b_W}) \tag{1}$$

with $V \in \mathbb{R}^{h \times m}$ and $\mathbf{b_V} \in \mathbb{R}^h$ denoting the weight matrix and the bias of a perceptron unit, whereas $W \in \mathbb{R}^{h \times c}$ and $\mathbf{b_W} \in \mathbb{R}^c$ are the parameters of the neuron performing classification over the $c$ target classes.

## 2.2 Q-MTL ARCHITECTURE

The Q-MTL network behaves similarly to the model introduced in Section 2.1, with the notable exception that it trains $k$ distinct classification models, all of which operate over the same hidden representation as input obtained from a single bi-LSTM unit.

More concretely, we replace the single prediction of the STL model from Eq. 1 by a series of predictions for Q-MTL according to

$$\mathbf{y_{i,j}} = softmax(ReLU(V^{(j)}\mathbf{h_i} + \mathbf{b_V^{(j)}})W^{(j)} + \mathbf{b_W^{(j)}}), \qquad (2)$$

with $j \in \{1, \ldots, k\}$. As argued before, this approach behaves favorably from a computational point of view, as it relies on a shared representation $\mathbf{h_i}$ for all the $k$ classification units.

The loss of the network for token position $i$ and gold standard class label $\mathbf{y_i^*}$ can be conveniently generalized as

$$l_{Q-MTL}(i) = \sum_{j=1}^{k} CE(\mathbf{y_i^*}, \mathbf{y_{i,j}}),$$

where $CE$ denotes categorical cross entropy loss and $k$ is the number of (identical) tasks in the Q-MTL model, with the special case of $k = 1$ resulting in standard single task learning (STL).

Losses from the different outputs can be aggregated efficiently during backpropagation, hence the shared LSTM cell benefit from multiple error signals without the actual need of going through multiple individual forward and backward passes.

Q-MTL outputs $k$ predictions by all of its prediction units, however, we can as well derive a combined prediction from the distinct outputs of Q-MTL according to

$$\frac{1}{k}\sum_{j=1}^{k} softmax(ReLU(V^{(j)}\mathbf{h_i} + \mathbf{b_V^{(j)}})W^{(j)} + \mathbf{b_W^{(j)}}), \qquad (3)$$

which essentially is a weighted average according to the predicted probabilities of the distinct models. As introducing averaging at the model-level would eliminate diversity of the individual classifiers (Lee et al., 2015), this kind of averaging took place in a post-hoc manner, only when making predictions.

## 2.3 TRADITIONAL ENSEMBLE MODEL

As an additional model, we also employ a traditional ensemble of $k$ independently trained STL models. We define the prediction of the ensemble model by averaging the predictions of $k$ independent models as

$$\frac{1}{k}\sum_{j=1}^{k} softmax(ReLU(V^{(j)}\mathbf{h_i^{(j)}} + \mathbf{b_V^{(j)}})W^{(j)} + \mathbf{b_W^{(j)}}). \qquad (4)$$

The distinctive difference between Eq. 4 and the Q-MTL model formulation in Eq. 3 is that ensembling relies on the hidden representations originating from $k$ independently trained LSTM models as denoted by the superscripts of the hidden states in $\mathbf{h_i^{(j)}}$. Such an ensemble necessarily requires approximately $k$-times as much computational resources compared to Q-MTL, due to the LSTM models being trained in total isolation. For the above reason, ensembling is a strictly more expensive form of training a model, for which reason we regard its performance as a glass ceiling for Q-MTL.

## 3 EXPERIMENTS

Our model uses character embeddings of 100 dimensions and the word representations get initialized by the 64-dimensional pre-trained polyglot word embeddings (Al-Rfou et al., 2013) as suggested by Plank & Agić (2018). We use a one-layered bi-LSTM which outputs hidden vectors $\mathbf{h_i} \in \mathbb{R}^{200}$ as a concatenation of $\overrightarrow{\mathbf{h_i}}, \overleftarrow{\mathbf{h_i}} \in \mathbb{R}^{100}$. Instead of directly applying a fully-connected layer to perform classification based on $\mathbf{h_i}$, we first transform $\mathbf{h_i}$ by an intermediate perceptron unit with ReLU

Table 1: Statistics on training data size.

|  | el | en | eu | fi | hr | hu | id | nl | ta | tr |
|---|---|---|---|---|---|---|---|---|---|---|
| # sentences | 1662 | 2738 | 5369 | 14980 | 6983 | 910 | 4477 | 12269 | 400 | 3685 |
| # word forms | 9035 | 7436 | 19222 | 39717 | 33382 | 7767 | 19223 | 26665 | 2637 | 13781 |
| # total words | 42326 | 50096 | 72974 | 127602 | 154055 | 20166 | 97531 | 186046 | 6329 | 38082 |

Table 2: Results of Q-MTL on the dev sets for varying number of tasks employed ($k$).

| $k$ | el | en | eu | fi | hr | hu | id | nl | ta | tr | Avg. |
|---|---|---|---|---|---|---|---|---|---|---|---|
| 1 | 95.61 | 94.99 | 94.49 | 93.19 | 96.84 | 93.95 | 93.05 | 96.05 | 82.74 | 93.61 | 93.45 |
| 10 | 95.84 | **95.23** | **94.81** | **93.30** | **96.99** | **94.25** | 92.98 | **96.53** | **84.48** | **93.78** | **93.82** |
| 30 | **95.86** | 95.21 | 94.59 | 93.09 | 96.93 | 93.79 | **93.25** | 96.27 | 83.85 | 93.46 | 93.63 |

activation. The perceptron transforms $\mathbf{h_i}$ into 20 dimensions, that is, we have $V \in \mathbb{R}^{20 \times 200}$. Our motivation with the extra non-linearity introduced by ReLU is to encourage an increased diversity in the behavior of the different output units.

Upon training the LSTMs, we used the default architectural settings employed by Plank et al. (2016), i.e., we relied on a word dropout rate of 0.25 (Kiperwasser & Goldberg, 2016) and an additive Gaussian noise (with $\sigma = 0.2$) over the input embeddings. We trained all our models for 20 epochs using stochastic gradient descent with a batch size of 1. First, we assess the quality of Q-MTL towards POS tagging, then we evaluate it on named entity recognition as well.

When comparing the performance of different approaches, Q-MTL models are compared against the average performance of $k$ STL models, where $k$ denotes the number of task in the case of Q-MTL. The $k$ STL models are also used to derive a single prediction by the ensemble model.

### 3.1 POS TAGGING EXPERIMENTS

We set our POS tagging related experiments on 10 treebanks from the Universal Dependencies dataset v2.2 (Nivre et al., 2018), namely the Greek-GDT (el), English-LinES (en), Basque-BDT (eu), Finnish-FTB (fi), Croatian-SET (hr), Hungarian-Szeged (hu), Indonesian-GSD (id), Dutch-Alpino (nl), Tamil-TTB (ta) and Turkish-IMST (tr) treebanks. These treebanks not only cover a typologically diverse set of languages, but they also vary substantially in the number of available training sequences, as illustrated in Table 1. Table 1 also illustrates the typological diversity of the investigated languages, as the average number of occurrences per distinct word forms vary substantially, i.e., between 2.4 for Tamil and 6.9 for Dutch.

#### 3.1.1 EXPERIMENTS WITH THE NUMBER OF TASKS

We first investigate how does changing the value for $k$, i.e., the number of simultaneously learned tasks, affects the performance of Q-MTL. We experimented with $k \in \{1, 10, 30\}$. Based on the results in Table 2, we set the number of tasks to be employed as $k = 10$ for all upcoming experiments. In order to choose $k$ without overfitting to the training data, this experiment was conducted on the development set.

#### 3.1.2 COMPARING Q-MTL WITH STL

Following the recommendation in (Dodge et al., 2019), we report learning curves over the development set as a function of the number epochs in Figure 2 As a general observation, we can see that Q-MTL tends to perform consistently better than STL models right from the beginning of training.

**Directly comparing the classifiers** One benefit of Q-MTL is that it learns $k$ different classification models during training with only a marginal computational overhead compared to training a STL baseline, since all the tasks share a common internal representation. As discussed earlier, we can combine the predictions from the $k$ classifiers from Q-MTL according to Eq. 2. It is also possible, however, to use the $k$ distinct predictions of Q-MTL. In what follows next, we compare the

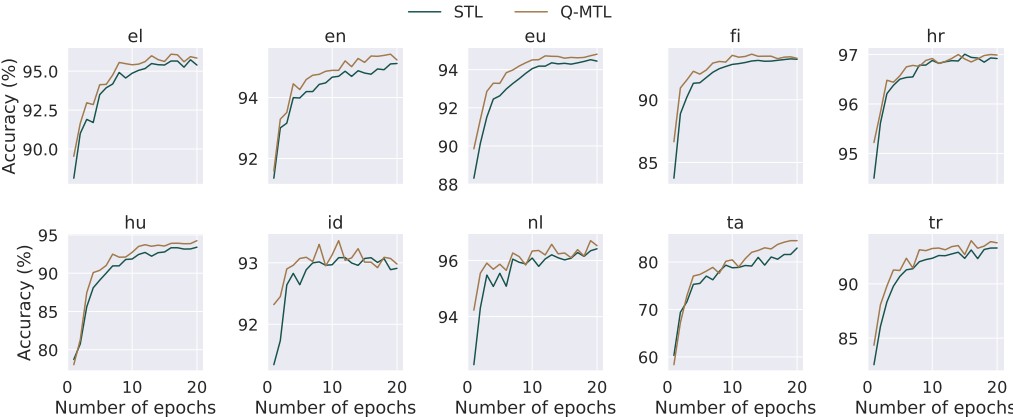

Figure 2: The accuracy of the different model types over the training epochs on the dev set.

performance of the $k$ STL models we train to the $k$ classifiers that are incorporated within a Q-MTL model.

Upon comparing the performance of a Q-MTL classifier with a STL model, we made it sure that the overlapping parameters (matrices $V$ and $W$) were initialized with the same values and that they receive the training instances in the exact same order. This way the performance achieved by the $i^{th}$ output of Q-MTL is directly comparable with the $i^{th}$ STL baseline. Comparison of the results of the individual outputs of Q-MTL and their corresponding STL counterpart are included in Figure 3.

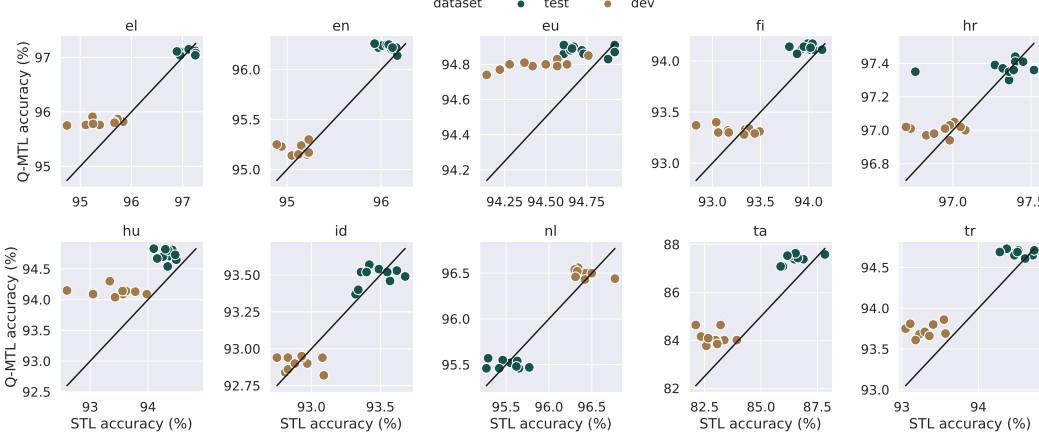

Figure 3: Scatter plot comparing the accuracy of the individual classifiers from Q-MTL ($k = 10$) and their corresponding STL counterpart. Each model that is above the diagonal line performs better after training in the Q-MTL setting.

Training Q-MTL models with $k$ tasks simultaneously is not only faster than training $k$ distinct STL models separately, but the individual Q-MTL models typically outperform their baseline counterparts evaluated against both the development and the test data.

**The regularizing effect of Q-MTL** We have argued earlier that Q-MTL has an implicit regularizing effect. Among most recent techniques, such as dropout (Srivastava et al., 2014), weight decay (Krogh & Hertz, 1992) is one of the most typical form of regularization for fostering the generalization capability of the learned models. When employing weight decay, we add an extra term penalizing the magnitude of the values learned by our model, which results in an overall shrinkage in the values of the model parameters.

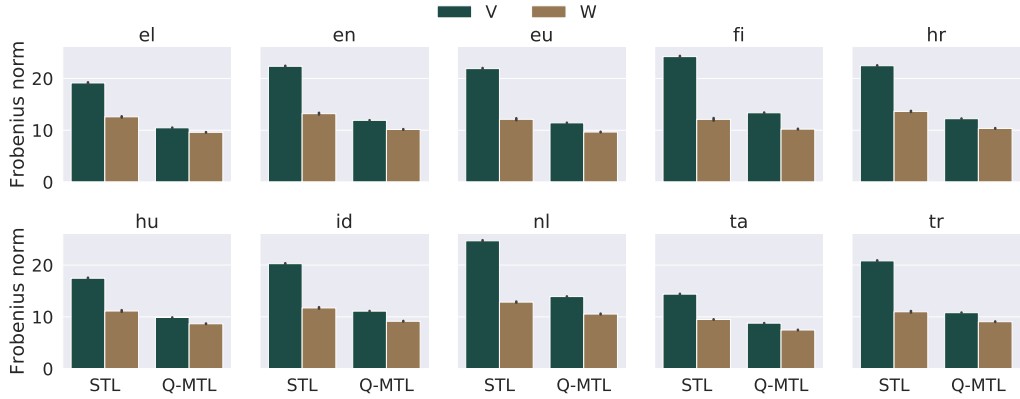

Figure 4: The average Frobenius norms of the learned parameter matrices $V$ and $W$ for the different approaches and treebanks.

Figure 4 illustrates that the effects of employing Q-MTL is similar to applying weight decay, as the Frobenius norm of the parameter matrices from the classifiers of Q-MTL are substantially smaller than those of the STL classifiers. This observation holds for both the of parameter sets $V$ and $W$. Recall that the initial values for these matrices were identical for both Q-MTL and STL.

### 3.1.3 COMPARISON TO AN ENSEMBLE OF CLASSIFIERS

We have provided a detailed comparison of the STL and Q-MTL models so far. We next extend their comparative evaluation with the ensemble model. Upon providing a comparison for the different approaches, we also assess their sensitivity towards the presence of noisily labeled tokens during training. To do so, we conducted multiple experiments for each language, for which we randomly replaced the true class label of a token by some predefined probability $p \in \{0, 0.1, 0.2, 0.3\}$. During the random replacement of the class labels, we ensured that the same tokens got randomly relabeled by the same label for the different approaches.

Figure 5 contains the performance of the three different models in conjunction with the different amounts of noisy labels introduced to the training set. We can observe from Figure 5 that Q-MTL outperforms STL for all the languages irrespective to the amount of noisy tokens being present encountered during training. Figure 5 further reveals that the performances of the ensemble models – which are based on the predictions of the STL classifiers – are dominantly better than the average performance of the individual STL models. When mislabeled tokens are not present in the training data at all, ensemble also has a slight advantage over Q-MLT, however, this advantage of the ensembling model gradually fades out as the proportion of noisy training labels increases. Indeed, for the case when 30% of the training labels are randomly replaced, the performance of Q-MTL reaches that of the ensemble model (cf. the rightmost subplot in Figure 5). The Q-MTL approach has the additional benefit over the ensemble model that it requires a fraction of computational resources as we will demonstrate it in Section 3.2.

As a final interesting note, the Q-MTL has an improved performance for Indonesian as the amount of noisy training labels increases. A possible explanation for this is that corrupting the class labels of the training data can be viewed as an alternative form of label smoothing (Szegedy et al., 2016), which is known to increase the generalization ability of neural models.

### 3.2 COMPARISON OF TRAINING TIMES

One of the main benefits of Q-MTL resides in its training efficiency compared to traditional ensemble models as also demonstrated by Figure 6, which includes the training times for the different approaches. We plot the training times on the logarithmic scale for better readability for both $k = 10$

and $k = 30$. We can see that the training times for STL and Q-MTL practically concur, whereas the overall costs of ensembling exceeds the training time of STL and Q-MTL models by a factor of $k$.

The training times reported in Figure 6 were obtained without GPU acceleration – on an Intel Xeon E7-4820 CPU – in order to simulate a setting with limited computational resources. We also repeated training on a TITAN Xp GPU. The GPU-based training was 3 to 10 times quicker depending on the languages, but the relative performance between the different approaches remained the same, i.e., STL and Q-MTL training times did not differ substantially, whereas the ensemble model took $k$-times as much time to be created.

### 3.3 EVALUATION ON NAMED ENTITY RECOGNITION

We also conducted experiments on the CoNLL 2002/2003 shared task data on named entity recognition (NER) in English, Spanish and Dutch (Tjong Kim Sang, 2002; Tjong Kim Sang & De Meulder, 2003). For these experiments, we report performance in terms of overall F1 scores calculated by the official scorer of the shared task. We trained models with $k = 10$ and compared the average performance of the individual STL models to the performance of the Q-MTL and ensemble models.

Table 4a shows the results for NER over the different languages, corroborating our previous observation that Q-MTL is capable of closing the gap between the performance of STL models and the much more resource-intensive ensemble model derived from $k$ independent models.

In our POS tagging experiment, we trained models on treebanks of radically differing sizes (cf. Table 1), whereas during our NER experiments, we had access to training data sets of comparable sizes (ranging between 218K and 273K tokens). In order to simulate the effects of having access to limited training data on NER as well, we artificially relied on only 10% of the available training sets.

These results for the limited training data setting are included in Table 4b, from which we can see that Q-MTL manages to preserve a larger fraction of its original performance, i.e., 87.5% on average as opposed to the ensemble and STL models, which preserved only 86.7% and 86.4% of their original F-scores, respectively.

## 4 COMPARISON TO PSEUDO-TASK AUGMENTATION

An achitecture introduced by Meyerson & Miikkulainen (2018) called Pseudo-Task Augmentation (PTA) uses a similar architecture to leverage a better representation of the task by fitting multiple outputs to the same task. This architecture makes a series of predictions according to

$$\mathbf{y_{i,j}} = softmax(W^{(j)}\mathbf{h_i} + \mathbf{b_W^{(j)}}). \tag{5}$$

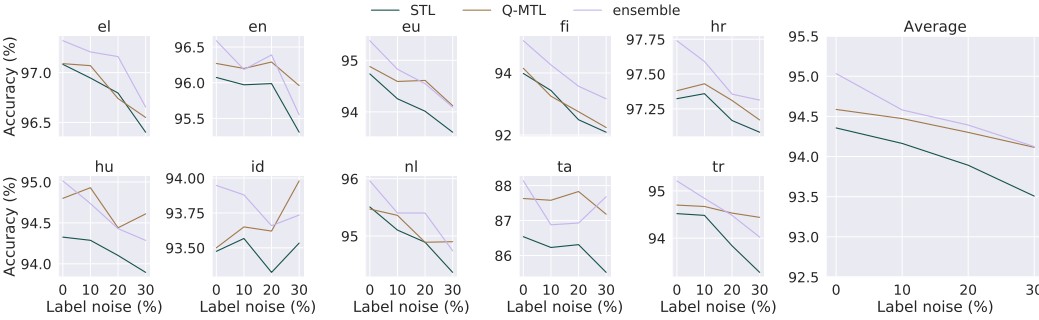

Figure 5: Model performances obtained by the different approaches when a varying amount of noisy training samples are introduced during training. The rightmost plot (titled Average) contains the averaged accuracies over the 10 treebanks.

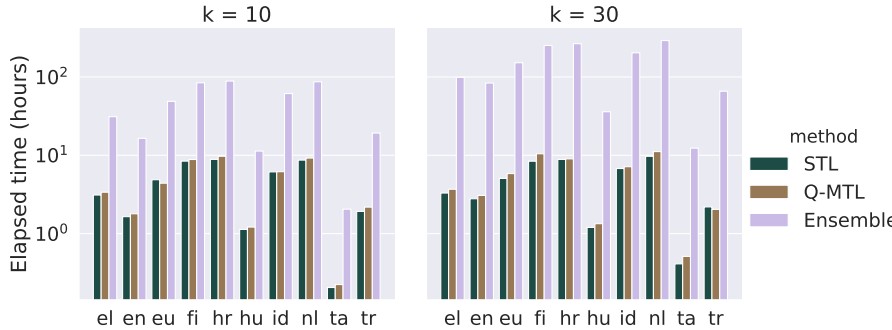

Figure 6: Training times of the different approaches for the different languages.

Table 3: F1 performance scores for the NER experiments.

| | Avg. STL | Q-MTL | Ensemble | | Avg. STL | Q-MTL | Ensemble |
|---|---|---|---|---|---|---|---|
| en | 86.68 | 86.88 | **87.86** | en | 77.54 | **80.24** | 78.52 |
| es | 82.28 | 82.35 | **83.76** | es | 70.71 | 71.57 | **72.56** |
| nl | 81.84 | 83.15 | **83.61** | nl | 68.47 | 69.16 | **70.33** |
| Avg. | 83.60 | 84.13 | **85.07** | Avg. | 72.42 | 73.66 | **73.80** |
| | (a) 100% training data used | | | | (b) 10% training data used | | |

The main difference between PTA and Q-MTL is that Q-MTL uses an extra transformation and a ReLU non-linearity over the hidden representation of the LSTM (cf. Eq. 2 and Eq. 5 for Q-MTL and PTA, respectively).

Another key difference is that PTA uses model selection (MS), whereas Q-MTL relies on model averaging (MA). This means that PTA makes prediction for test instances during inference based on the model which achieves best performing dev set accuracy. Q-MTL, however, aggregates all the models according to 3.

As Figure 7 shows the different settings for model selection and Multi-Layered Perceptron (MLP) that explore the methods used in PTA (MS @ 0 MLP) to Q-MTL (MA @ 20 MLP). We also we evaluated the different combinations of model selection/averaging and the usage of additional non-linearity for completeness. In these experiments 20 MLP means the MLP has a 20 dimensional

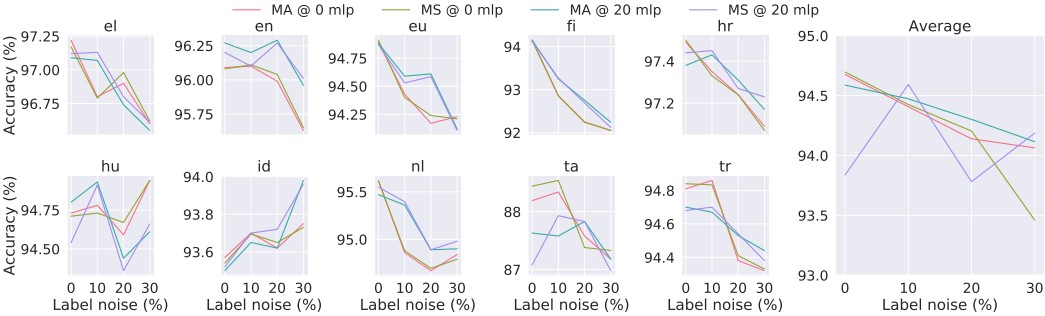

Figure 7: PTA and Q-MTL compared in an analogue manner. Model selection (MS) is compared to model averaging (MA) and MLP (20 MLP) compared to linear classifier (0 MLP). From PTA (MS @ 0 MLP) to Q-MTL (MA @ 20 MLP) we can see all combinations of these parameters.

output, while 0 MLP means there is no MLP layer added to the tasks. Note, that by removing the MLP layer we also remove the ReLU activation.

Figure 7 demonstrates that the Q-MTL model with its MLP layer can facilitate the use of model averaging shown in Eq. 3 as it outperforms the Q-MTL using model selection (MS @ 20 MLP). On the other hand in the case no MLP is used (0 MLP) with which it is indeed discouraged to ensemble the models, we can see that MS often outperforms MA in case of 0 MLP. Additionally, we can see that the MLP layer improves the tolerance of the models to the increasing label noise, as it outperforms 7 out of 10 treebanks the model not employing extra non-linearity. Interestingly when the train set contains high label noise, the ensemble of linear classifiers outperforms the model selected ones.

## 5 RELATED WORK

Caruana (1997) showed that neural networks can be trained for multiple tasks, leveraging cross domain information. More recently, Søgaard & Goldberg (2016); Sanh et al. (2018) argues that solving low-level NLP tasks can improve the performance of high level tasks. Additionally, Plank et al. (2016); Bingel & Søgaard (2017) show that better performing models can be trained by introducing multiple auxiliary tasks. Rei (2017) proposes an auxiliary task for NLP sequence labeling tasks, where the auxiliary tasks is to predict the previous and next word in the sequence. Our results complement these findings by showing that better generalization can also be achieved if we learn multiple models for the same task concurrently.

Meyerson & Miikkulainen (2018) introduced Pseudo-Task Augmentation a similar architecture that aims to build a robust internal representation from multiple classifier units that optimize for the same task in the same neural network. Their work contains policies that simulate linear independence (refered as: perturb, hyperturb, independent dropout) in the set of linearly dependent classifier units, while making use of the dependence in other cases to improve accuracy (refered as: greedy). Section 4 describes the similarities and differences to our method. PTA architecture is evaluated on multitask as well, while our work only considers single tasks at the moment.

Ruder & Plank (2018) has shown that self-learning and tri-training can be adapted to deep neural nets in the semi-supervised regime. Their tri-training architecture resembles our approach in that they were utilizing multiple classifier units that were built on top of a common representation layer for providing labels to previously unlabeled data.

Cross-view training (CVT) (Clark et al., 2018) resembles Q-MTL in that it also employs a shared bi-LSTM layer used by multiple output layers. The main difference between CVT and Q-MTL is that we are utilizing an bi-LSTM to solve the same task multiple times in a supervised setting, whereas Clark et al. (2018) used it to solve different tasks in a semi-supervised scenario.

A series of studies have made use of ensemble learning in the context of deep learning (Hansen & Salamon, 1990; Krogh & Vedelsby, 1995; Lee et al., 2015; Huang et al., 2017). Our proposed model is also related to the line of research on mixture of experts proposed by Jacobs et al. (1991), which has already been applied successfully in NLP before (Le et al., 2016). The main difference in our proposed architecture is that the internal LSTM representation that we train are shared across the classifiers, hence a more efficient training could be achieved as opposed to training multiple independent expert models as it was done in (Shazeer et al., 2017).

Model distillation (Hinton et al., 2015) is an alternative approach for making computationally demanding models more effective during inference, however, the approach still requires training of a "cumbersome" model first.

## 6 CONCLUSIONS

We proposed quasi-multitask learning (Q-MTL), which can be viewed as an efficiently trainable alternative of traditional ensembles. We additionally demonstrated that it acts as an implicit form of regularization as well. In our experiments, Q-MTL consistently outperformed the single task learning (STL) baseline for both POS tagging and NER. We have also illustrated that Q-MTL generalizes better on smaller and noisy datasets compared to both STL and ensemble models.

The computational overhead for the additional classification units in Q-MTL is infinitesimal due to the effective aggregation of the losses and the shared recurrent unit between the identical tasks. Although we evaluated our approach over sequence classification tasks, the general idea can be applied for other network architectures and beyond NLP applications as well.

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
