# OpenReview forum: "Frustratingly easy quasi-multitask learning"
_ICLR.cc/2020/Conference — Reject_

### Official Review · AnonReviewer3 · 2019-10-23
**Official Blind Review #3**

**Rating:** 1

**Review:**

This paper considers a regularization technique, derived from multi-task learning, where multiple models with some shared parameters are jointly trained to solve copies of the same task. The technique is well-motivated as an efficient alternative to ensemble learning. The method is validated for a BiLSTM NLP model, which is applied to several POS tagging and named entity recognition tasks. The power of the technique as a regularizer is also demonstrated in the case of highly noisy labels, surprisingly, even outperforming ensemble learning in this setting.

Despite this, my inclination is to reject the paper because of the substantial overlap with previous work and the limited scope of experiments.

My primary concern is that the proposed technique was previously introduced in [1]. This prior work is not acknowledged in the current paper. Perhaps it was overlooked because it is situated tightly in Multi-task Learning, whereas the present work is motivated mainly with respect to Ensembling.

Although the general method was introduced previously, the paper does have some key experimental differences that would be interesting to see explored further.

(1)	The paper uses a hidden layer in the separate classification heads, whereas previous work only used a linear classifier. The intuition that more complex heads will yield more diverse models is clear, but it would be great to see experimental evidence that this complexity helps. The conclusion states that the computational overhead is “infinitesimal”; does increasing the complexity of the classifier trade cost for performance?
(2)	This paper uses Eq. 3 to make predictions, whereas previous work found that this did not improve over simply using the best single prediction model, which makes prediction somewhat more efficient. Is there some experimental evidence that Eq. 3 leads to improvements?
(3)	This paper considers the comparison to ensembling, whereas previous work only considered comparisons to single task and standard multitask learning. Additional experiments showing the advantages over ensembling could make this extension a significant contribution.
(4)	This paper presents novel investigation of the regularization effects of the method, i.e., the resilience to noisy labels and the analysis of learned weight matrices. Is there a real problem where this resilience to noise will improve over ensembles, i.e., without randomly replacing labels? Such an experiment would make this point more compelling. Also, is there some underlying reason why the method outperforms ensembles in this case? Is it simply because the method is less expressive so cannot overfit?

In effect, if the paper could clearly show that (1) or other practical extensions lead to improvements over ensembling in settings where ensembling is commonly used, or enable ensembling in settings where vanilla ensembling fails (i.e., the case of noisy labels), then it could be a substantial contribution. The current scope of the experiments is too limited to conclusively show these points. For example, the technique can be applied to any architecture, but the experiments in the paper are limited to a single architecture; and additional experiments with architectures and tasks that commonly use ensembling would make the experiments more compelling, ideally with comparisons to external results.

Other minor comments:
-	It would be good to see the number of model parameters for easy comparison, especially in table 2 with different value of k.
-	It looks like the x and y axis labels are swapped in Figure 3; from the Figure it looks like STL gets higher accuracies.
-	Figure 2 should say epochs instead of iterations.

[1] Meyerson, E. & Miikkulainen R. “Pseudo-task Augmentation: From Deep Multitask Learning to Intratask Sharing---and Back”, ICML 2018.


**Experience Assessment:**

I have published in this field for several years.

**Review Assessment: Checking Correctness Of Derivations And Theory:**

I carefully checked the derivations and theory.

**Review Assessment: Checking Correctness Of Experiments:**

I assessed the sensibility of the experiments.

**Review Assessment: Thoroughness In Paper Reading:**

I read the paper thoroughly.

---

> ### Author Response · Authors · 2019-11-12
> **Response to Review #3**
>
> First, we would like to thank the reviewer for the valuable feedback on the submission.
>
> Admittedly, we were not aware of the recent independent line of research from Meyerson and Miikkulainen.
> We are very grateful to the review for drawing our attention to this publication.
>
> After reading the suggested ICML paper which introduces pseudo-task augmentation (PTA), we agree with the reviewer that what we propose in our paper is indeed very similar -- although not entirely identical -- to PTA. We also share that view of the reviewer that we can potentially deliver certain interesting complementary results to the PTA approach. Having became aware of PTA, we shall definitely reframe and back up with additional experiment certain claims in our paper.
> In the upcoming days, we are planning to update the paper taking into consideration the suggestions and remarks of the reviewer that we found indeed very inspiring and helpful.

---

### Official Review · AnonReviewer2 · 2019-10-28
**Official Blind Review #2**

**Rating:** 3

**Review:**

This paper proposes a quasi-multitask learning (Q-MTL) for supervised learning. The network architecture in Q-MTL borrows the idea of multi-task neural networks by sharing the latent representation among different classifiers which are designed for a single task.

What is the difference among multiple classifiers for the single task? Can they become identical? The rationale behind Q-MTL is unclear to me. Authors need to conduct more analyses to show why Q-MTL is superior to supervised learning.

Authors claim that Q-MTL is equivalent to performing some regularization. However, I did not see any analysis on this aspect.

In experiments, the performance of Q-MTL is not so good when compared with ensemble learning.

**Experience Assessment:**

I have published in this field for several years.

**Review Assessment: Checking Correctness Of Derivations And Theory:**

I assessed the sensibility of the derivations and theory.

**Review Assessment: Checking Correctness Of Experiments:**

I assessed the sensibility of the experiments.

**Review Assessment: Thoroughness In Paper Reading:**

I read the paper at least twice and used my best judgement in assessing the paper.

---

> ### Author Response · Authors · 2019-11-12
> **Response to Review #2**
>
> We would like to thank R2 for the time spent reviewing the paper. Below we comment on the questions (Q) and remarks (R) included in the review.
>
> Q: What is the difference among multiple classifiers for the single task? Can they become identical?
> A: They are initialized with different weight matrices and we employ an extra ReLU nonlinearity to encourage their diversity, i.e. preventing the different decoders converging to the same set of parameters. In principle, the different decoders _could_ become identical, however, we have not experienced this to happen in our experiments (as a consequence of the measures we took for avoiding such an unwelcome outcome).
>
> R: The rationale behind Q-MTL is unclear to me.
> A: The basic assumption is that forcing the LSTM to develop such an internal representation which can serve as a useful input to multiple classifiers simultaneously helps it to obtain a more robust representation.
>
> R: Authors need to conduct more analyses to show why Q-MTL is superior to supervised learning.
> A: We have shown that Q-MTL consistently performs better than single task learning (STL). It is true that ensemble models typically performed better than Q-MTL, however, the computation need for constructing an ensemble classifier is orders of magnitude higher than that of Q-MTL. Indeed, the central question of our research was if it is possible to ''enjoy the benefits of ensemble learning, while avoiding its overhead for training models from scratch multiple times?''
>
> R: Authors claim that Q-MTL is equivalent to performing some regularization. However, I did not see any analysis on this aspect.
> A: Please, refer to the paragraph within Section 3.1.2 which is entitled "The regularizing effect of Q-MTL". Here we demonstrated that Q-MTL serves as an implicit form of weight decay regularization. Additionally, the fact that Q-MTL is more tolerant towards the presence of corrputed training instances with noisy labels implies that it can generalize better compared to the alternative approaches we experimented with. We believe that this improved generalization property of Q-MTL originates from the implicit regularization it performs.
>
> R: In experiments, the performance of Q-MTL is not so good when compared with ensemble learning.
> A: Please note that training an ensemble classifier requires orders of magnitudes more computational power, hence ''we regard its performance as a glass ceiling for Q-MTL''. We do show that Q-MTL has favorable performance over STL, with STL requiring a comparable computational budget to Q-MTL.

---

### Decision · Program_Chairs · 2019-12-19

**Decision:**

Reject

**Comment:**

One of the reviewers pointed out similarity to existing very recent work which would require significant reframing of the current paper. Hence, this work is below the bar at the moment.